

# Functional histology of the skin in the subterranean African giant mole-rat: thermal windows are determined solely by pelage characteristics

Lucie Pleštilová[1], Jan Okrouhlík[1], Hynek Burda[2], Hana Sehadová[3,4], Eva M. Valesky[5] and Radim Šumbera[1]

[1] Department of Zoology, Faculty of Science, University of South Bohemia, České Budějovice, Czech Republic
[2] Department of Game Management and Wildlife Biology, Faculty of Forestry and Wood Sciences, Czech University of Life Sciences, Prague, Czech Republic
[3] Department of Molecular Biology and Genetics, Faculty of Science, University of South Bohemia, České Budějovice, Czech Republic
[4] Institute of Entomology, Biology Centre of the Czech Academy of Sciences, České Budějovice, Czech Republic
[5] Department of Dermatology, Venereology and Allergology, University Hospital, Johann Wolfgang Goethe Universität Frankfurt am Main, Frankfurt am Main, Germany

Corresponding author
Radim Šumbera, sumbera@prf.jcu.cz

## ABSTRACT

Excavation of burrows is an extremely physically demanding activity producing a large amount of metabolic heat. Dissipation of its surplus is crucial to avoid the risk of overheating, but in subterranean mammals it is complicated due to the absence of notable body extremities and high humidity in their burrows. IR-thermography in a previous study on two species of African mole-rats revealed that body heat was dissipated mainly through the ventral body part, which is notably less furred. Here, we analyzed the dorsal and ventral skin morphology, to test if dermal characteristics could contribute to higher heat dissipation through the ventral body part. The thickness of the epidermis and dermis and the presence, extent and connectivity of fat tissue in the dermis were examined using routine histological methods, while vascular density was evaluated using fluorescent dye and confocal microscopy in the giant mole-rat *Fukomys mechowii*. As in other hitherto studied subterranean mammals, no subcutaneous adipose tissue was found. All examined skin characteristics were very similar for both dorsal and ventral regions: relative content of adipose tissue in the dermis (14.4 ± 3.7% dorsally and 11.0 ± 4.0% ventrally), connectivity of dermal fat (98.5 ± 2.8% and 95.5 ± 6.8%), vascular density (26.5 ± 3.3% and 22.7 ± 2.3%). Absence of large differences in measured characteristics between particular body regions indicates that the thermal windows are determined mainly by the pelage characteristics.

## INTRODUCTION

Mammals are able to maintain a stable and relatively high body temperature in a wide range of ambient temperatures, which is achieved by heat production and heat loss regulation (*McNab, 2002*; *Withers et al., 2016*). When mammals perform energy consuming activities, production of body heat increases substantially, yet surplus heat would cause overheating and has to be dissipated (*McNab, 2002*; *Schmidt-Nielsen, 1997*; *Sherwood, Klandorf & Yancey, 2013*). Mammals can lose heat by physical routes that is radiation, convection, conduction, or evaporation. Evaporation is the most effective way of cooling; however, it is limited in water-saturated environments or when water for sweating is not available (*Baldo, Antenucci & Luna, 2015*; *McNab, 2002*; *Withers et al., 2016*).

Heat dissipation in mammals can be enhanced in body areas known as thermal windows (*Feldhamer et al., 2015*; *Withers et al., 2016*). These areas are usually sparsely haired and situated at body appendages as pinnae in elephants or rabbits (*Mohler & Heath, 1988*; *Weissenböck et al., 2010*), tail in coypus and beavers (*Krattenmacher & Rübsamen, 1987*; *Steen & Steen, 1965*), or feet in foxes or otters (*Klir & Heath, 1994*; *Kuhn & Meyer, 2009*). Thermal windows are usually well vascularized with numerous arteriovenous anastomoses, regulating the heat transfer by vasodilatation and vasoconstriction (*Bryden & Molyneux, 1978*; *Khamas et al., 2012*; *Vanhoutte et al., 2002*). There are, for example, two main blood vessel plexuses beneath the dark patches of giraffe skin considered as thermal windows, which facilitate heat exchange with the environment (*Ackerman, 1976*; *Mitchell & Skinner, 2004*). The blood vessel walls are also thinner in the patches than in non-patch lighter skin (*Mitchell & Skinner, 2004*). In the proximal region of the wing of the Brazilian free-tailed bat *Tadarida brasiliensis*, a network of arteries and veins positioned perpendicular to the body has been found, which is unique among bats and which is considered to be a thermal window allowing effective thermoregulation during migration (*Reichard et al., 2010*).

Due to its low heat conductivity, a fat layer can be an important component of heat conservation, particularly for aquatic mammals living in cold water (*Bryden, 1964*; *Kvadsheim & Folkow, 1997*; *Liwanag et al., 2012*). These mammals have a thick continuous insulative subcutaneous fat layer all over the body, except for the extremities used for active thermoregulation (*Khamas et al., 2012*; *Schmidt-Nielsen, 1997*). The insulative properties of fat layer have also been proven in much smaller laboratory mice (*Kasza et al., 2014*, *2016*).

Heat dissipation is particularly challenging in burrowing mammals. Digging in a mechanically resistant substrate is energetically demanding as it produces a lot of metabolic heat (*Ebensperger & Bozinovic, 2000*; *Lovegrove, 1989*; *Luna & Antenucci, 2007*; *Zelová et al., 2010*). We estimate that subterranean mammals (i.e., burrowing mammals, which forage underground) spend several hours per day by digging burrows and transporting excavated soil. Such activity inevitably produces a surplus of metabolic heat. However, the burrow atmosphere is typically very humid (with relative humidity often exceeding 90%) (reviewed in *Burda, Šumbera & Begall (2007)*), which impairs the efficacy

of evaporative cooling (*Šumbera, 2019*). Moreover, subterranean mammals usually lack longer body appendages, which would facilitate heat radiation (see above). The ability to lose heat via convection in sealed tunnels with very restricted (if any) air currents is also extremely limited (*Baldo, Antenucci & Luna, 2015*; *Burda, Šumbera & Begall, 2007*). The best way to dissipate a surplus of heat seems to be cooling via conduction through appressing the body surface to the colder tunnel walls. Indeed, relatively high thermal conductance in subterranean rodents suggests this way of cooling (*Buffenstein, 2000*).

It was speculated that in subterranean mammals, the ventral body surface is relevant for heat dissipation as indicated by its shorter and less dense fur (*Cutrera & Antenucci, 2004*; *Šumbera et al., 2007*). In two species of African mole-rats (Bathyergidae), the silvery mole-rat *Heliophobius argenteocinereus* and the giant mole-rat *Fukomys mechowii*, the importance of the less furred ventral body part as the main thermal window was supported also by infrared thermography (*Šumbera et al., 2007*; *Okrouhlík et al., 2015*). Recently, the higher surface temperature of the ventral body part in a wide gradient of experimental ambient temperatures was confirmed in other species of subterranean rodents from different phylogenetic lineages (F. Vejmělka and R. Šumbera, 2017, unpublished data).

The morphology of thermal windows in subterranean mammals is heavily understudied. *Šumbera et al. (2007)* found notable differences in pelage between the dorsal and ventral body parts in two African mole-rat species. The pelage was four times sparser on the belly than on the back in the giant mole-rat and even nine times sparser in the silvery mole-rat. In the latter species, hairs on the ventral body part were also shorter. Similar differences between ventral and dorsal body regions were found in 15 species of subterranean rodents of different phylogenetic lineages (F. Vejmělka and R. Šumbera, 2015, unpublished data). The importance of fur for heat dissipation has been demonstrated in the South African highveld mole-rat *Cryptomys hottentotus pretoriae* experimentally, with fur shaving decreasing body temperature, probably as a result of increased heat dissipation (*Boyles et al., 2012*). This finding indicates that fur characteristics are probably highly relevant for heat dissipation in subterranean species.

Skin morphology of the African mole-rats is rather understudied and attention has mainly been paid to the hairless skin of the naked mole-rat *Heterocephalus glaber* (*Daly & Buffenstein, 1998*; *Sokolov, 1982*; *Thigpen, 1940*; *Tucker, 1981*). *Daly & Buffenstein (1998)* found a dense capillary network in the superficial layers of the dermis in the naked mole-rat. *Kimani (2013)* provided a detailed description of skin morphology of different body regions in *H. glaber* and the African root-rat *Tachyoryctes*. He noticed that the skin on the dorsal side is thicker than on the ventral side in both species and explained it by higher resistance to wear during digging. However, while the thickness of skin on the ventral side is comparable in both species, it is much thicker in *Tachyoryctes* on the dorsal side. This difference in the thickness can be caused by presence of fat in the hypodermis on dorsal side of *Tachyoryctes*, which can affect thermoregulation.

In this study, we focus on skin characteristics of the social *F. mechowii*, a species in which the role of ventral body size in heat dissipation has been suggested on the grounds of its higher surface temperatures and lower pelage insulation (*Šumbera et al., 2007*;

*Okrouhlík et al., 2015*). We focus on a comparison of the ventral body part (where higher heat dissipation is expected) with the dorsal body part (where heat dissipation should be limited due to the isolating effect of denser fur). There is a question, of whether heat dissipation on the ventral body part is facilitated due to poor thermal insulation of fur only, or whether the characteristics of the skin also contribute to heat dissipation. If characteristics of the skin on the ventral body part are relevant for heat dissipation, we should expect lower fat content and connectivity allowing heat exchange through gaps in an insulative layer of fat, and also higher vascularization enabling higher heat transport as described in other mammals (*Ackerman, 1976*; *Atlee et al., 1997*; *Khamas et al., 2012*; *Mitchell & Skinner, 2004*; *Reichard et al., 2010*).

# MATERIALS AND METHODS

## Study animals

The giant mole-rat *Fukomys mechowii* is the largest social African mole-rat with a body mass of 200–600 g (*Kawalika & Burda, 2007*). It is distributed in the Democratic Republic of Congo, Zambia and Angola, where it inhabits mesic savannas, forests, bushlands and agricultural fields (*Kawalika & Burda, 2007*; *Wilson, Lacher & Mittermeier, 2017*). Animals involved in our study were born in captivity and housed at a temperature of 25 ± 1 °C.

We examined skin samples (from the dorsal and ventral sides) of five adult non-breeding females of age 2.5–10.5 years and body weight 186–308 g. The samples were taken with a biopsy needle (6 mm diameter) from two freshly thawed cadavers stored under −20 °C and from three perfused specimens (see below). The latter animals were sacrificed in the framework of neuroanatomical studies (cf. *Kverková et al., 2018*). All five specimens are stored at the Faculty of Sciences, University of South Bohemia, under the accession numbers 2296, 7940, 8280, 9330, 9653.

## Histology

Epidermal and dermal thickness as well as thickness, proportion, connectivity and pattern of dermal fat were assessed in skin samples from all five specimens. From each individual, six skin samples were taken from the dorsal and six from the ventral body part (see Fig. 1 for details of sampling points).

Three perfused animals were first processed for analysis of vascularization (see below) and then they were processed by routine histological methods. The two frozen specimens were studied by routine histological methods.

Biopsy samples were fixed in 4% buffered paraformaldehyde (PFA, pH = 7.2), dehydrated by ascending concentrations of ethanol in automatic tissue processor (Leica ASP200S) and embedded in paraffin wax in tissue embedder (Leica EG1150H). Paraffin blocks were sectioned on a rotary microtome (Leica RM2255) to obtain skin cross-sections, which were then mounted on a glass slide and stained with hematoxylin and eosin by an autostainer (Leica XL ST5010). The sections were examined under a light microscope (Olympus CX41) with 20× objective magnification and photographed with a digital camera (Olympus DP74).

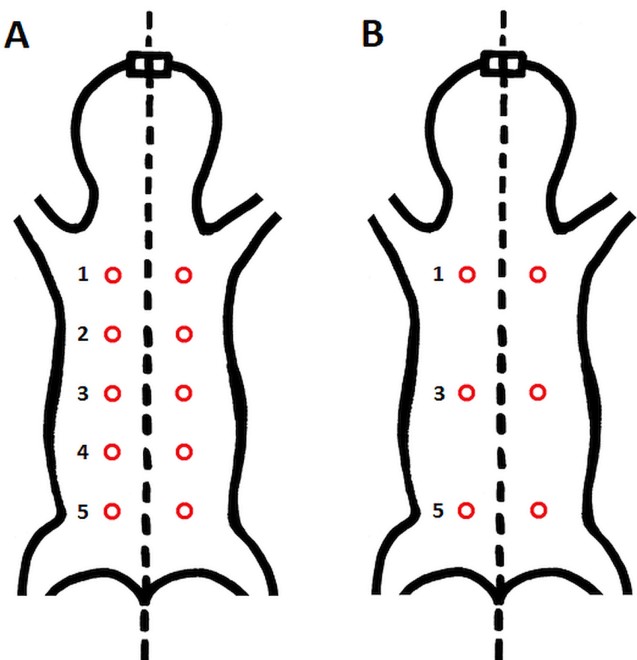

**Figure 1 Schematic location of skin sampling points on the animal.** (A) Sampling points for evaluation of the area covered by vessels; (B) sampling points for histological evaluation of epidermis, dermis and fat tissue characteristics. Samples were taken from both dorsal and ventral body side.

The micrographs were processed using ImageJ 1.48v (*Schindelin et al., 2012*). A square grid of side length 300 μm was randomly overlaid over the entire micrograph. Thickness of epidermis, dermis and dermal fat was measured on 12 random points and respective mean values were calculated. Thickness of epidermis was measured in the direction perpendicular to its border with dermis. Thickness of dermis and fat layer thickness were measured on the internal–external axis of the animal. Dermis was measured from the epidermal junction to the border of the epimysium (Fig. 2).

The extent, connectivity and pattern of the dermal fat were established as follows. A square grid of side length 50 μm was overlaid above the section micrograph. The presence of adipocytes within each grid cell throughout the section was determined and used to calculate the extent of dermal fat as a proportion of grid cells with adipocytes present to the total number of grid cells of the section. To evaluate the dermal fat layer connectivity and pattern we defined seven categories of its thickness based on the total thickness along the internal–external axis—0, 1–50, 51–100, 100–150, 150–200, 200–250 and >250 μm. In each section, total width of each thickness category on axis parallel with animal surface was measured and divided by the width of the section to calculate the percentage of the dermal fat thickness categories. We defined fat layer connectivity as a proportion of the width of the section containing fat tissue to the total width of the section and we used the fat layer thickness categories as a measure of adipose tissue pattern.

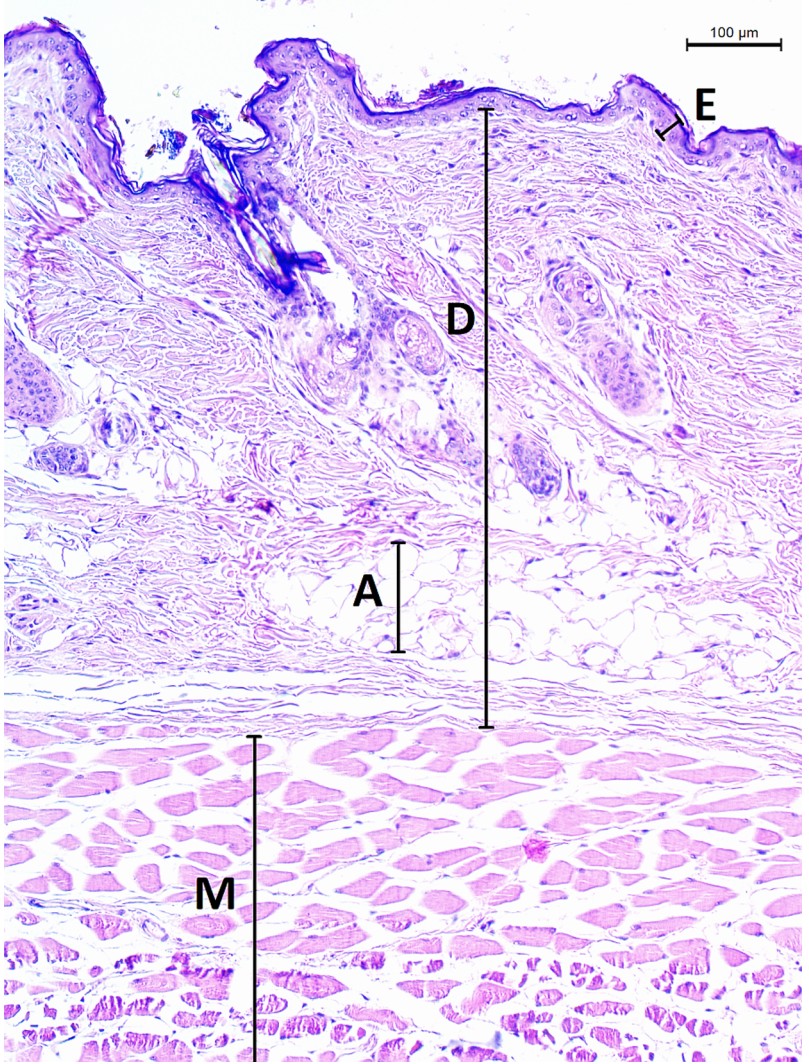

**Figure 2 Histological section of skin in *Fukomys mechowii*.** The section of thickness 4 μm was stained by hematoxylin and eosin, D, dermis; E, epidermis; M, muscle layer; A, adipose tissue.

## Vascularization density

The vessel system of three females was stained with fluorescent dye by transcardial perfusion by DiI as described by *Li et al. (2008)* and studied by confocal microscopy. In short, animals were perfused with 300 ml of heparinized PBS (pH 7.4) followed by perfusion by 120 ml of lipophilic carbocyanine dye (solution of 1,1′-dioctadecyl-3,3,3′,3′-tetramethylindocarbocyanine perchlorate). Afterwards, the vascular system was perfused with 300 ml 4% buffered PFA. On the next day, skin samples were taken by a biopsy needle and stored in 1% buffered PFA. There were 20 samples per animal in total (see Fig. 1 for details about sampling points). Skin samples were thoroughly rinsed with PBS (four times for 15 min at RT and overnight at 4 °C) and transferred to 100% glycerol through glycerol series (30%, 50%, 80% for 30 min each), mounted on microscope slides and examined under confocal laser scanning microscope (Olympus FV3000)

using an objective with 4× magnification (Olympus UPlanSApo4x). Pin-hole size was set automatically to 140 μm, optical filters were set to match ALEXA 568 (i.e., excitation wavelength 561 nm, emission wavelength 603 nm, detection wavelength 570–620 nm) and grayscale color depth was 12 bit. Other microscope parameters were as follows: laser transmissivity 0.30%, PMT voltage 350 V, gain 1×, offset 2%, sampling speed 12.5 μs/pixel and resolution 1024 × 1024. Samples were optically sectioned in the $Z$-axis in 17–20 planes with 30 μm distance. Four tiles per plane were imaged and the whole skin sample image in each plane was reconstructed by software stitching (Olympus FV31). To obtain a single plane image with projection of all blood vessels, maximal $Z$ projection of all planes was then performed (ImageJ 1.48v). Contrast of the resulting image was then enhanced using The Curves tool in Gimp GNU GPL v3+ (*Solomon, 2009*) so that all vessels were clearly visible. For further processing the image color resolution was then reduced to 8-bit. Vessel area and density was then established in the software AngioTool (*Zudaire et al., 2011*). Vessel density was calculated as proportion (in %) of vessel area to whole area of the sample.

As we did not compare the results statistically, due to the low number of examined individuals, mean values ± SD are given throughout the text and tables unless stated otherwise.

## RESULTS

The epidermis of *F. mechowii* consists of 3–10 cell layers covered by a corneous layer. The dermis consists of dense irregular connective tissue with more fibroblasts in the papillary layer and a regular arrangement of hair follicles in the reticular layer (Figs. 2 and 3). The subcutaneous layer (hypodermis) is not clearly delimitated and consists of sparse connective tissue, which is frequently thin and lacks adipocytes. Adipocytes within the dermis are rarely present singularly; they usually form clusters of different sizes and they frequently surrounds growing hair follicles (Fig. 3).

The thickness of the epidermis, dermis and dermal fat layer for the dorsal and ventral skin of each of five specimens are given in Table 1. Mean thickness of the epidermis was higher on the ventral side in most specimens, but the differences between specimens and also between measurements within one side were higher than differences between the dorsal and ventral sides. The difference between mean thickness of the dermis and fat layers on both body sides was negligible, whereas the individual variability was relatively high. The pattern of dermal fat was variable between specimens as well and it did not show any obvious trend (Fig. 4). The relative content of adipose tissue in the dermis was 14.4 ± 3.7% on the dorsal and 11.0 ± 4.0% on the ventral side (Table 1) and the dermal fat layer connectivity was 98.5 ± 2.8% on the dorsal and 95.5 ± 6.8% on the ventral side (Fig. 5). The most represented fat layer thickness category was 100–150 μm and the majority of the area was occupied by a fat layer of thickness 50–200 μm. Detailed proportions of fat layer thickness categories in the dermis of each mole-rat is given in Table S1.

Mean vessel density counted as a proportion of the area occupied by vessels inside the maximal projection of whole the explant area (Fig. 6) was 24.6 ± 3.4% ($n = 3$) and it was

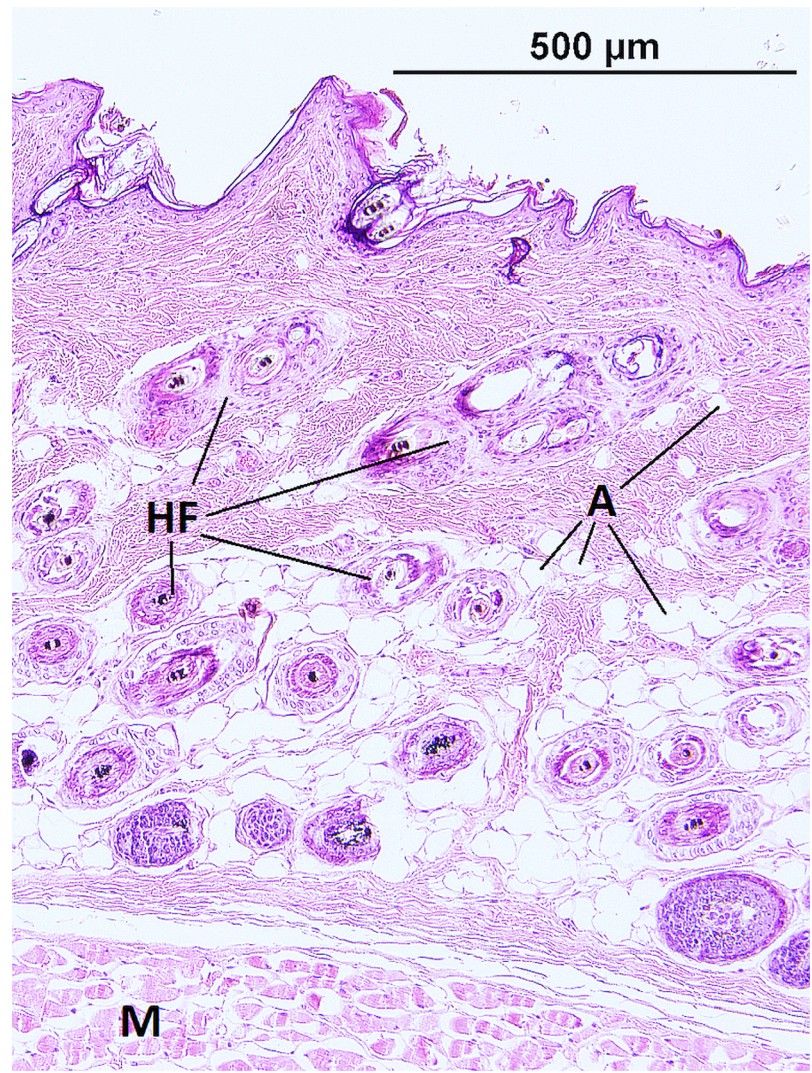

**Figure 3 Histological section of skin in *Fukomys mechowii* showing distribution of adipocytes surrounding hair follicles.** The section of thickness 4 μm was stained by hematoxylin and eosin. A, adipocytes; HF, hair follicles with associated glands; M, muscle.

slightly higher on the ventral than on the dorsal side, 26.5 ± 3.3% and 22.7 ± 2.3% (*n* = 3), respectively (Fig. 7). Detailed information about each sample is given in Table S2.

## DISCUSSION

We described skin characteristics such as the thickness of epidermis and dermis, the relative extent and pattern of distribution of white adipose tissue in the dermis and the vessel density (vascularization) on the dorsal and ventral sides of the trunk in a strictly subterranean rodent, the giant mole-rat. All these characteristics were expected to play a role in the dissipation of metabolic heat, especially after energy consuming burrowing, which is a typical activity of subterranean mammals. Since we postulate the existence of thermal windows enhancing heat dissipation on the ventral surface of mole-rats,

**Table 1  Skin characteristics on dorsal and ventral body part in five females of *Fukomys mechowii*.** Mean ± SD of epidermis, dermis and fat tissue thickness and proportion of fat tissue in dermis for dorsal and ventral side of each animal.

| Animal ID | Body mass (g) | Epidermis (µm) | | Dermis (mm) | | Fat layer (mm) | | Fat (%) | |
|---|---|---|---|---|---|---|---|---|---|
| | | Dorsum | Venter | Dorsum | Venter | Dorsum | Venter | Dorsum | Venter |
| 2296 | 187 | 19.4 ± 6.0 | 21.6 ± 7.5 | 0.7 ± 0.2 | 0.9 ± 0.2 | 0.1 ± 0.1 | 0.2 ± 0.1 | 14.3 ± 2.8 | 12.1 ± 3.3 |
| 7940 | 186 | 20.3 ± 8.2 | 19.6 ± 8.6 | 1.0 ± 0.2 | 1.0 ± 0.3 | 0.2 ± 0.1 | 0.1 ± 0.1 | 10.8 ± 3.1 | 8.1 ± 3.8 |
| 8280 | 308 | 13.9 ± 4.6 | 18.4 ± 6.3 | 0.8 ± 0.2 | 0.8 ± 0.3 | 0.1 ± 0.0 | 0.1 ± 0.1 | 14.1 ± 1.8 | 11.2 ± 2.5 |
| 9330 | 301 | 20.9 ± 6.3 | 21.4 ± 6.4 | 1.2 ± 0.1 | 1.3 ± 0.3 | 0.3 ± 0.1 | 0.2 ±0.1 | 15.0 ± 3.5 | 9.6 ± 1.3 |
| 9653 | 290 | 10.8 ± 4.8 | 14.0 ± 5.9 | 0.7 ± 0.2 | 0.7 ± 0.2 | 0.2 ± 0.1 | 0.2 ± 0.1 | 17.8 ± 4.2 | 14.1 ± 4.6 |

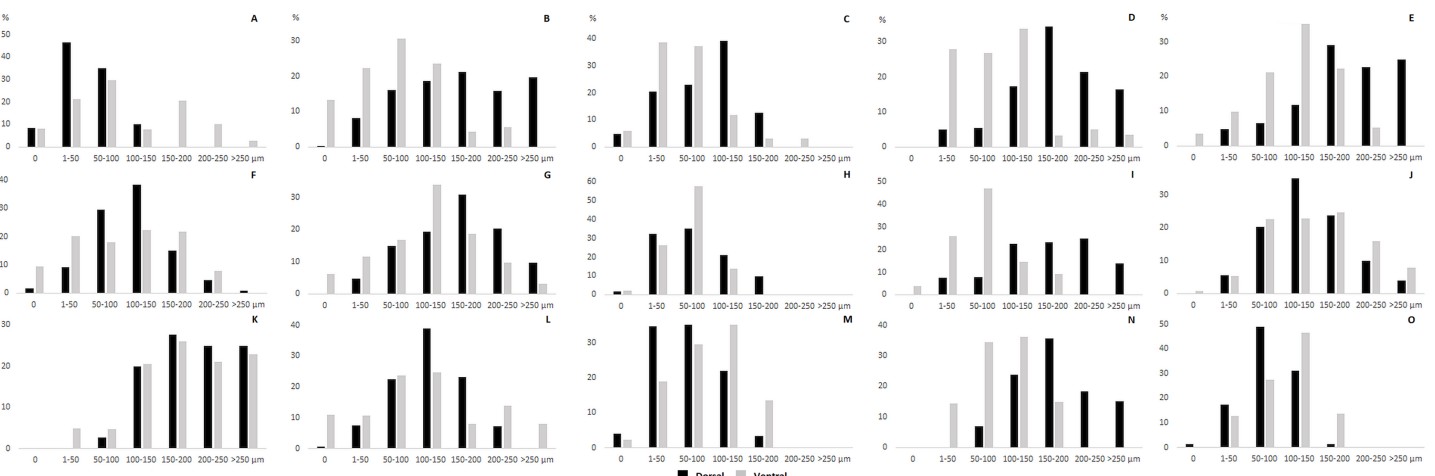

**Figure 4  Dermal fat tissue pattern in *Fukomys mechowii*.** Mean percentage occupied by dermal fat thickness categories in different body parts of each individual: 2296 (A) anterior, (F) middle, (K) posterior, 7940 (B) anterior, (G) middle, (L) posterior, 8280 (C) anterior, (H) middle, (M) posterior, 9330 (D) anterior, (I) middle, (N) posterior, 9653 (E) anterior, (J) middle, (O) posterior. Dorsal body part is in black, ventral body part is in grey.               

we expected to find morphological differences of the skin between the ventral and dorsal body parts. Surprisingly, we did not find any notable differences between the two body surfaces. Although we studied only a few individuals (note that the evaluation of vessel density requires the sacrifice of living animals), the study still provides good insight into the skin structure and its role in heat dissipation in subterranean rodents, a topic that has not been studied so far.

The thickness of epidermis and dermis of the giant mole-rat was within the range found by *Sokolov (1982)* for different rodent taxa including fossorial ones, that is, 8–80 µm and 0.2–2.8 mm respectively. *Sokolov (1982)* speculated that scratch diggers have thicker skin on the breast while chisel-tooth diggers have thicker skin on the back, as mechanical protection against wear during digging. Our findings did not confirm this idea, at least for the species in our study, a chisel digger.

Dermal adipose tissue is well developed in many mammalian species and, in rodents, it is clearly separated from the subcutaneous fat tissue by panniculus carnosus (*Driskell et al., 2014*; *Wojciechowicz et al., 2013*). Although the dermal fat tissue plays a role

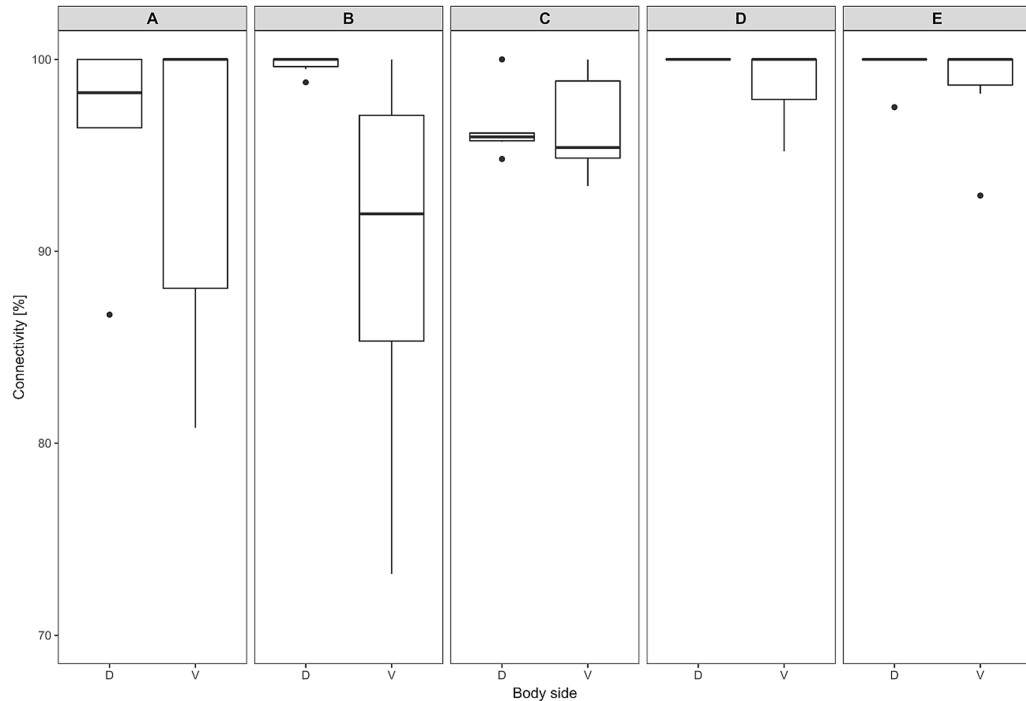

**Figure 5 Dermal fat tissue connectivity in *Fukomys mechowii*.** Proportion of skin sample width containing any dermal fat tissue. Boxplots showing medians (horizontal lines), quartiles (boxes), 5 and 95 percentiles (whiskers) and outliers (black dots) for each animal. Letters (A–E) denote each specimen (ID numbers 2296, 7940, 8280, 9330, 9653, respectively), D, dorsal and V, ventral body side.

in protection and regeneration of skin and hair growth cycle (reviewed in *Alexander et al. (2015)* and *Guerrero-Juarez & Plikus (2018)*), it also contributes to thermoregulation. The dermal fat layer was found to be thicker in mice facing chronic cold stress and it was calculated that just a 200 µm thick layer of dermal fat reduces heat loss by twofold in mice housed at an ambient temperature of 16 °C below the body temperature (*Kasza et al., 2014*, *2016*). Dermal adipose tissue forms a continuous layer in the laboratory mouse (*Kasza et al., 2014*) whereas in the naked mole-rat and in the common mole-rat, this tissue consists of either isolated or grouped adipocytes (*Daly & Buffenstein, 1998*). In the giant mole-rats, isolated adipocytes are present infrequently, as they form larger clusters situated relatively close to each other, as can be seen from the high connectivity of dermal fat tissue (Fig. 6). The shape of clusters of adipocytes surrounding the hair follicles can be influenced also by the hair cycle as is evident from the Figs. 2 and 3, and as was described and reviewed in *Guerrero-Juarez & Plikus (2018)*.

Blood vessels play a major role in heat exchange, especially in body parts that facilitate heat dissipation (*Bryden & Molyneux, 1978*; *Tarasoff & Fisher, 1970*). Our findings showed that relative vessel density is higher on the ventral than on the dorsal body part, yet the differences are not so prominent (less than 4%) as to cause significant differences in body heat dissipation between both body regions.

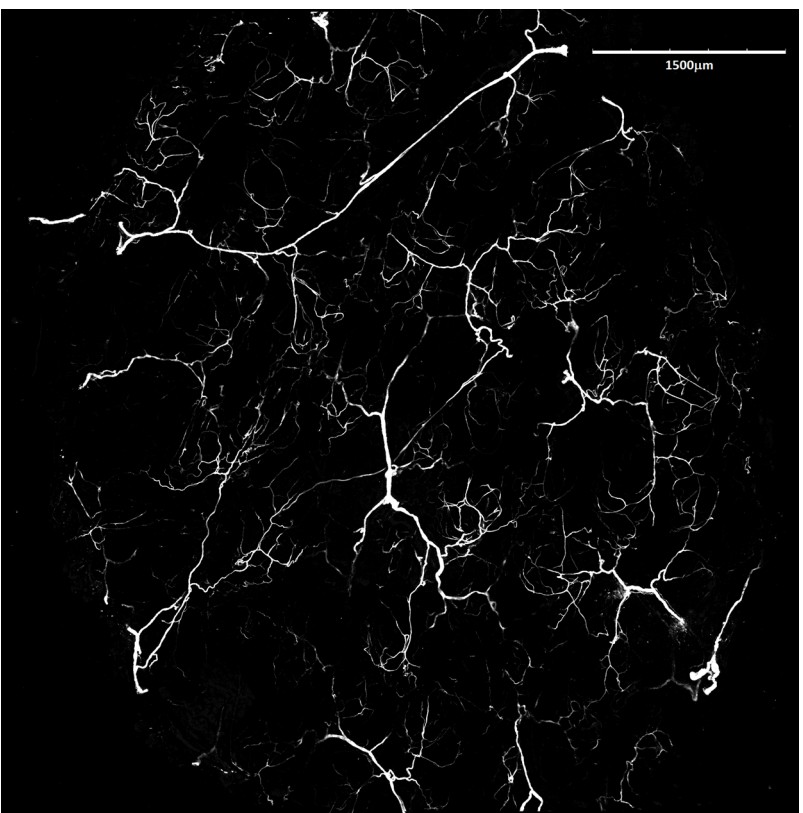

**Figure 6 Blood vessels of *Fukomys mechowii*.** Visualized by DiI labeling, enhanced maximal projection of 17 planes with 30 μm distance viewed by confocal microscope. Blood vessels are in white color.

In the hitherto studied African mole-rats, no continuous subcutaneous fat layer was found (*Daly & Buffenstein, 1998*; *Sokolov, 1982*), while in other mammals, typically, the hypodermis consists mainly of white adipose tissue (*Marquart-Elbaz et al., 2001*; *Martin et al. 2007*; *Matoltsy, 1986*; *Scudamore, 2014*; *Sokolov, 1982*). *Daly & Buffenstein (1998)* as well as *Sokolov (1982)* found only the aggregations of fat cells within the dermis of the naked mole-rat and common mole-rat, which is in agreement with our findings. Absence of subcutaneous fat tissue and thus the loose skin connection to the deeper fascia allows the integument slidability, which is known from human anatomy of skin in the eyelid or penis (*Van De Graaff, 2001*). The mobility of loose subcutaneous tissue in subterranean rodents protects skin from injuries (*Kawamata et al., 2003*), which can facilitate movement underground (*Daly & Buffenstein, 1998*). It is known that instead of storing fat in subcutaneous layer, at least some African mole-rats (and probably other subterranean mammals) deposit fat into the intraperitoneal cavity and around the neck (*O'Riain, Jarvis & Faulkes, 1996*; *Scantlebury, Speakman & Bennett, 2006*). Apart from this, the absence of a subcutaneous fat layer is highly relevant in thermal biology allowing fast dissipation of metabolic heat.

We found no notable differences in the content and connectivity of dermal fat and vascularization density between the ventral body part, that is, the area with expected heat

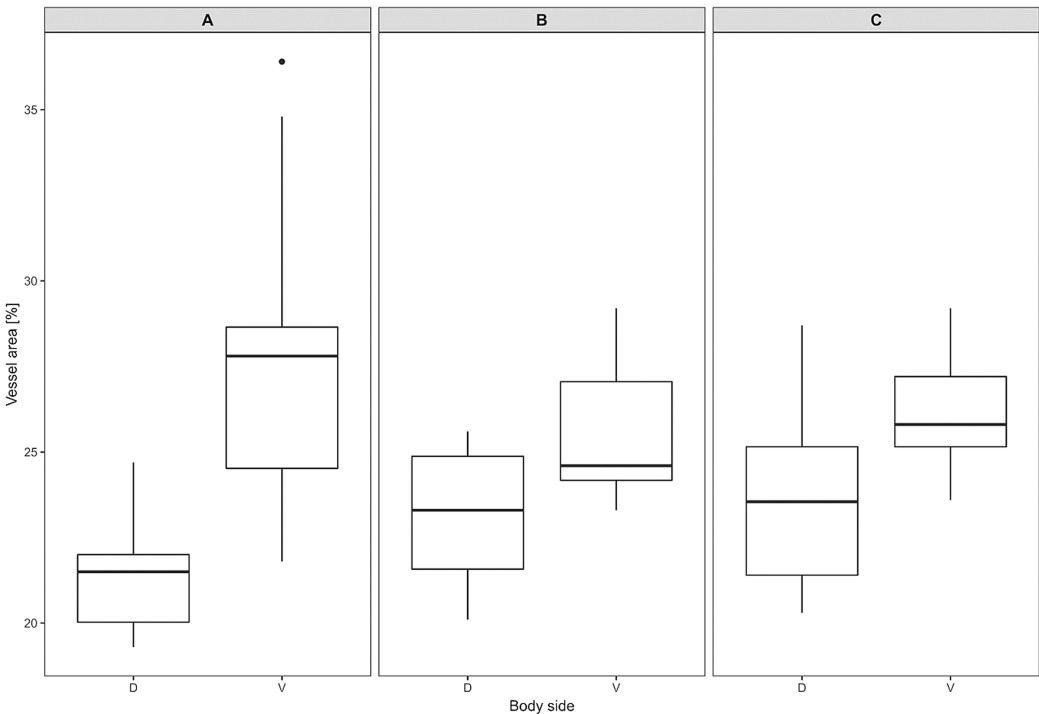

**Figure 7 Vascular density in *Fukomys mechowii*.** Percentage of area covered by vessels in maximal projection of whole skin sample. Boxplots showing medians (horizontal lines), quartiles (boxes), 5 and 95 percentiles (whiskers) and outlier (black dot) in each animal. (A–C) Denote specimens 8280, 9330, 9653, respectively, D, dorsal and V, ventral body side.               

dissipation function in subterranean rodents (cf. *Šumbera et al., 2007*; *Cutrera & Antenucci, 2004*) and the dorsal body part. This is in contrast with a prominent difference in the insulative characteristics of fur (hair length and density) between both body areas. In the giant mole-rat, dorsal pelage is four times denser (having the same length), which must certainly contribute substantially to heat conservation (*Šumbera et al., 2007*). The important role of fur in mole-rat thermoregulation is indicated also by experiments with fur shaving in Mashona mole-rats *Fukomys darlingi* and highveld mole-rats *Cryptomys hottentotus pretoriae* (*Boyles et al., 2012*). In mole-rats, heat dissipation could be easily influenced by different patterns of fur characteristics across the body together with some behavioral patterns such as curling up and thus hiding the ventral, less furred area under cold temperatures. Seasonal changes of microenvironmental conditions in burrows could be mitigated by seasonal moulting, which is known in some bathyergids (*Hislop & Buffenstein, 1994*).

## CONCLUSIONS

If we consider all findings of the present study about insulative value of skin and vessel density together with the findings on fur (*Šumbera et al., 2007*), we may conclude that pelage characteristics are probably the most important factor for dissipation or conservation of body heat in the giant mole-rat. For future studies, a focus on the potential

role of hair reduction on the feet or even the short tail on heat dissipation could be interesting. It is known that feet can also contribute notably to heat dissipation, so histological analysis of the pads could bring some new information on this issue.

## ACKNOWLEDGEMENTS

We would like to thank Kristina Kverková and Pavel Němec for sharing the perfunded specimens, Jitka Pflegrová for preparing samples and Matěj Lövy for disscussion. We also thank to Monika Doll, Katja Haerle and Jutta Mueller for help with processing of samples.

### Funding

This work was supported by The Czech Science Foundation 31-17-19896S. The funders had no role in study design, data collection and analysis, decision to publish, or preparation of the manuscript.

### Grant Disclosures

The following grant information was disclosed by the authors:
The Czech Science Foundation: 31-17-19896S.

### Competing Interests

The authors declare that they have no competing interests.

### Author Contributions

- Lucie Pleštilová performed the experiments, analyzed the data, prepared figures and/or tables, authored or reviewed drafts of the paper, and approved the final draft.
- Jan Okrouhlík conceived and designed the experiments, performed the experiments, analyzed the data, prepared figures and/or tables, authored or reviewed drafts of the paper, and approved the final draft.
- Hynek Burda conceived and designed the experiments, authored or reviewed drafts of the paper, and approved the final draft.
- Hana Sehadová performed the experiments, analyzed the data, authored or reviewed drafts of the paper, and approved the final draft.
- Eva M. Valesky performed the experiments, authored or reviewed drafts of the paper, and approved the final draft.
- Radim Šumbera conceived and designed the experiments, prepared figures and/or tables, authored or reviewed drafts of the paper, and approved the final draft.

### Data Availability

Raw data are available in Tables S1 and S2. All five specimens are stored at the Faculty of Sciences, University of South Bohemia: 2296, 7940, 8280, 9330, 9653.

## Supplemental Information

Supplemental information for this article can be found online at http://dx.doi.org/10.7717/peerj.8883#supplemental-information.

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
