# Peer review of "Functional histology of the skin in the subterranean African giant mole-rat: thermal windows are determined solely by pelage characteristics"

_PeerJ, doi:10.7717/peerj.8883_

## Round 0.1 · original submission · Major Revisions

Please find attached comments on your manuscript from two reviewers. They both have raised important points to address in your manuscript. In addition, I have some points of my own.

Your results section is very short. Whilst I appreciate its concision, I think in some instances you need to describe the results you obtained in a bit more detail. For instance, with regard to the thickness of the epidermis, the dermis, and the dermal fat layer, the reader is simply referred to table 1. You need to summarise the results in the text – were the dorsal layers thicker or less thick than the ventral layers, or was there no difference? Similarly, the results of the fat layer thickness analysis are given one sentence and all the data is in a supplementary table. Is there no way of presenting the results of this analysis graphically to give the reader a better understanding of the distribution of the fat layer?

My other main concern is where you have combined results from all three specimens to give a combined mean and combined standard deviation. Given that the sample values from within specimens are not independent, I am not convinced that this is statistically justifiable. And despite your claim not to be comparing the results statistically, by creating these grand means and grand standard deviations, you are implicitly making statistical comparisons. Given that you only have 3 specimens in each analysis, I think you can present the results specimen by specimen (as you have done in figures 4 and 6) without using combined means. In fact, I think it could be valid to statistically test for differences between the dorsal and ventral surfaces within specimens (as you have 6-10 data points in each), but not to use the data for all three specimens combined. I suggest removing the final line from Table 1 (I’m not sure how helpful the mean age or body mass of all specimens is anyway).

Otherwise, my points for revision are covered by the reviewers’ reports. Like reviewer 2, I was somewhat confused about the number of specimens – why were only 3/5 used for histological analysis? I also agree that the language needs correcting throughout the manuscript (in particular the use of articles). In this regard, please see the attached annotated manuscript, where I have endeavoured to correct all typographical and grammatical errors.

Lastly, please ignore reviewer 1’s comment about the references. PeerJ will make sure the references are correctly formatted later on in the publication process.

I look forward to seeing a revised version of the manuscript in due course.

Reviewer 1 ·

Basic reporting

no comments

Experimental design

no comments

Validity of the findings

no comments

Additional comments

Manuscript entitled “Functional histology of the skin in the subterranean African giant mole-rat: Thermal windows are determined solely by pelage characteristics” for PeerJ.
The authors have studied the comparison of ventral body part (where higher heat dissipation is expected) with the dorsal body side (where heat dissipation should be limited due to the isolating effect of denser fur) of the giant mole-rat Fukomys mechowii. For that, they analyzed the thickness of epidermis and dermis and presence, extent and connectivity of fat tissue in the dermis which were examined using routine histological methods, while vascular density was evaluated using fluorescent dye and confocal microscopy.
The paper is clear and well detailed. The purpose of the study is correctly defined. The details of most methods are comprehensible. The results are clearly presented. The discussion is relevant and complete. The conclusions are appropriately stated and connected to the original question investigated. The manuscript contains sufficient and proper references. The figures are good and help to analyze and comprehend significantly the results. Therefore I would suggest accepting the manuscript for publication in PeerJ after minor revision.
a) Materials and Methods.
It is not clear the use of “freshly thawed cadavers stored under -20 °C”. How do the authors freeze the specimens? How were they thawed? Do the authors use any cryoprotectants?
Freezing damages cell membranes, and reduces the histological readability of biological specimens. Slow freezing can cause distortion of tissue due to ice crystal formation that replaces the normal morphology.

It should be described the methodology used in detail.

b) The format of the references in the text should be checked and corrected. The use of and/& should be checked. Check the names of the journals, they must be changed to italic letter.

Reviewer 2 ·

Basic reporting

Subterranean rodents that excavate their burrows deal with the problem of heat dissipation in underground tunnels with high humidity. In a previous study the authors recognized that the giant mole-rat Fukomys mechowii uses mainly the sparsely haired ventral part of their body to dissipate heat. In the current study, the authors followed the question whether the animals show differences in their skin characteristics in the dorsal and ventral parts.
The manuscript is well written, but the text needs some language editing (especially the use of articles should be revised). I give some examples below, but it should be noted that I am not an English native speaker. The introduction provides sufficient background information to follow the authors’ rationale.

Experimental design

The research is original and falls within the aims and scope of PeerJ. The data presented is new. The research question is well defined and the methods have been described in detail, so that replication would be possible.
My major issues are the following:
1. The sample size is very low (n=5 for general description of skin histology; n=3 for studying the vascularization density) and information is partly confusing. Although the authors claim in the material & methods section that they have used five individuals for skin histology (L. 140-141), they present details of only three individuals (Figure 4, Table 1, Table S1) which I find highly confusing. Therefore, the authors should give the sample size for each mean value they present in the result section.
(further criticism see below)

Validity of the findings

1 (continued): Furthermore, the authors claim that a statistical analysis is not possible due to low sample size. However, without a proper statistical analysis their statements are not valid. The authors come for instance to the conclusion that the difference in vessel density between the ventral and the dorsal part are not so prominent (L. 264-265). But taking a look at Figure 6, invokes the impression that the difference is significant. Thus, the research question has not been solved. I suggest that the authors increase the sample size (at least for histology) in order to run statistical tests (no sacrifice is needed if you use frozen carcasses).
2. The labelling of Figure 2 is not complete and partly wrong. The measured thickness of the dermis seems to include the measured thickness of the epidermis, which is wrong! Did the authors measure the dermis as indicated? The authors should indicate the glands (also which type of glands?) and hairs. What is the extent of the subcutis? From the labelling one gets the impression that the muscle layer is directly beneath the dermis. Is that correct? Which histological textbook did the authors follow?

Additional comments

Here are some minor suggestions:
L30: “to test” sounds better than “to find”
L45: “If they perform energy consuming activity...” specify “they”
L49: I would use “ineffective” or “limited” instead of “restricted”
L59: Plural of “plexus” is “plexus” (not “plexuses”)
L64: “the bats” delete “the”
L76: “We may estimate” sounds a bit weird
L82: “facilitate the heat radiation” delete “the”
L85: “adpressing” never heard of this word before – is that correct?
L90: “of the African mole-rats” delete “the”
L92 and throughout the text (also in the figure labelling): “ventral body part” instead of “ventral body side” (the same for “dorsal body side”)
L99: “on the belly”
L100: “in the latter species” (not “later”)
L101: “on the ventral body part”
L103: “The importance”
L106: “probably highly relevant”
L110: Name the authors (and year) instead of “Later authors compared…”
L112: “the situation”
L115: “the thickness”
L118: “fat in the hypodermis” instead of “fat containing hypodermis” (the latter can be very misleading)
L125: “There is a question” is a weird expression
L168-169: “A square grid with a grid length of 50 µm…”
L174-178: This part of the method section is not very clear. Please rephrase.
L205: “the low number”
L209: “The dermis”
L211: “The subcutaneous layer”
L213: “within the dermis”
L217: What does “the most represented fat layer thickness category” mean?
L223: “is given in Table S2”
L227: “on the dorsal”
L230: “which is a typical activity”
L232: “in the skin”
L239: Which fossorial rodent species do you refer to?
L245: “mice and humans”
L253: “at an ambient temperature of”
L262: “play a major role”; “body parts that facilitate heat dissipation”
L272: “Absence of subcutaneous fat”
L277: “fat in this layer” specify which layer
L292: “across the body”
L298: “from the present study”

---

## Round 0.2 · accepted · Accept

Thank you for the revised version of your manuscript. I am satisfied that you have address all of my and the reviewers' comments and am happy to accept it for publication.

In particular, I am very pleased to see that you have added two extra specimens to your histological analysis as suggested by reviewer 2 – I think this really strengthens your results. Thank you for taking note of my requests to describe the results more fully in the text of the manuscript and to remove some of the summary statistics which rather implied statistical comparisons that you weren’t actually making.

Congratulations!